# TREATMENT EFFECT ESTIMATION WITH COLLIDER BIAS AND CONFOUNDING BIAS

## ABSTRACT

To answer causal questions from observational data, it is important to consider the mechanisms that determine which data values are observed and which are missing. Prior work has considered the treatment assignment mechanism and proposed methods to remove the confounding bias from the common causes of treatment and outcome. However, there are other issues in sample selection, commonly overlooked in prior work, that can bias the treatment effect estimation, such as the issue of censored outcome as a form of collider bias. In this paper, we propose the novel Selection Controlled CounterFactual Regression (SC-CFR) to simultaneously address confounding and collider bias. Specifically, we first calculate the magnitude of the collider bias of different instances by estimating the selection model and then add a control term to remove the collider bias while learning a balanced representation to remove the confounding bias when estimating the outcome model. Our theoretical analysis shows that we can achieve an unbiased treatment effect estimates from observational data with confounding and collider bias under certain assumptions. Extensive empirical results on both synthetic and real-world datasets show that our method consistently outperforms benchmarks on treatment effect estimation when both types of biases exist.

## 1 INTRODUCTION

Causal inference is a powerful statistical modeling tool for explanatory analysis and a central problem in causal inference is the estimation of treatment effect. The gold standard approach for treatment effect estimation is to conduct Randomized Controlled Trials (RCTs), but RCTs can be expensive (Kohavi & Longbotham, 2011) and sometimes infeasible (Bottou et al., 2013). Therefore, it is important to develop effective approaches to estimate treatment effect from observational data.

In observational studies, association does not imply causation, mainly due to the presence of (sample selection) biases in the data. There are two main sources of biases: confounding bias and collider bias (Hernán & Robins, 2020). To define confounding bias and collider bias, we use causal diagrams in Figure 1, and let $\mathbf{X}$ be the observed pre-treatment variables, $T$ be the treatment variable, and $Y$ be the outcome variable.

Confounding bias results from common causes of treatment and outcome (Guo et al., 2020; Greenland, 2003; Hernán & Robins, 2020). As shown in Figure 1(a), there are two sources of association between $T$ and $Y$: the path $T \to Y$ that represents the causal effect of $T$ on $Y$, and the path $T \leftarrow \mathbf{X} \to Y$ between $T$ and $Y$ that includes the common cause $\mathbf{X}$, named the backdoor path (Pearl, 2009), which introduces spurious associations into the observational data and results in $P(T \mid \mathbf{X}) \neq P(T)$. Confounding bias is very common in observational studies and can lead to incorrect treatment effect estimation. For example, when estimating the effect of job training programs on future incomes (LaLonde, 1986), work ability is a confounder that determines both whether an individual participated in the program and the individual's income. Due to the confounding bias, we may draw an incorrect conclusion about the effect of the job training programs on future incomes.

Collider bias is a special case of sample selection bias that results from conditioning on a common effect of $T$ and $Y$ (Greenland et al., 1999; Greenland, 2003; Hernan et al., 2004; Westreich, 2012; Elwert & Winship, 2014), as shown in Figure 1(b), where $S$ is the selection variable indicating whether a unit is selected, i.e., $S = 1$ when the unit is selected for observation and $Y$ is observable, otherwise $S = 0$ and we cannot observe $Y$ (Smith & Elkan, 2004b). Except for the path $T \to Y$, the

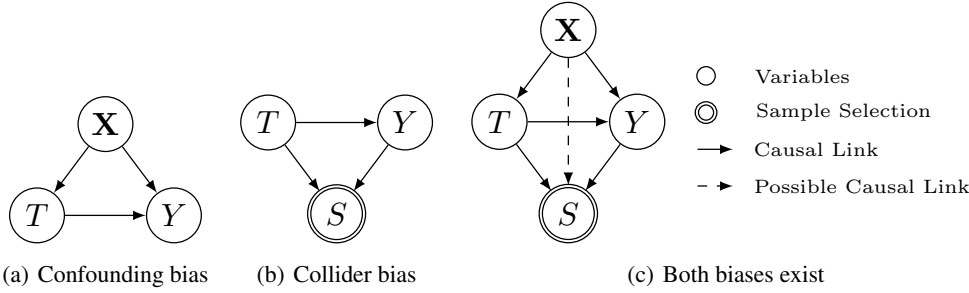

Figure 1: Causal diagrams with either confounding bias, collider bias or both.

other source of association between $T$ and $Y$ is from the open path $T \rightarrow S \leftarrow Y$ that links $T$ and $Y$ through their conditioned on common effect $S$, which results in $P(T, Y, \mathbf{X} \mid S = 1) \neq P(T, Y, \mathbf{X})$ and introduces spurious associations into the observational data. An analysis conditioned on $S$ will cause sample selection bias, i.e., we can only observe the outcome of those selected units, leading to incorrect treatment effect estimation (Westreich, 2012; Tattan-Birch et al., 2021).

Currently, many causal inference methods have been proposed to estimate treatment effect directly from observational data with confounding bias, including propensity score based methods (Rosenbaum & Rubin, 1983; Dehejia & Wahba, 2002; Hirano et al., 2003; Hirano & Imbens, 2004; Williamson et al., 2012), confounder balancing methods (Hainmueller, 2012; Kuang et al., 2017; Athey et al., 2018; Fong et al., 2018) and causal representation learning methods (Johansson et al., 2016; Shalit et al., 2017; Yao et al., 2018; Hassanpour & Greiner, 2020). However, existing causal inference works mostly ignore collider bias in data, and thus suffer from the most common case where confounding bias and collider bias are both present, as shown in Figure 1(c). In real-world scenarios, the above two biases both exist in observational data in most of time. Still taking the analysis of job training programs as an example, ones who has not participated in such programs with a lower income may be unwilling to report their current incomes, leading to collider bias. In this case, if we only control one of the two biases, the other will still affect our estimation. Therefore, it is necessary to develop an approach to solve both biases in treatment effect estimation.

Most of the previous work on selection bias (Heckman, 1979; Chib et al., 2009; Marchenko & Genton, 2012; Ding, 2014; Ogundimu & Hutton, 2016; Wiemann et al., 2022), including work that took confounding bias into account(Bareinboim et al., 2014; Bareinboim & Tian, 2015; Correa & Bareinboim, 2017), has only solved the simple case of selection bias caused by covariates or the treatment variable. However, for the more complex situation of collider bias, only (Bareinboim & Pearl, 2012) discussed the feasibility of removing it with the assistance of some variables that meet certain conditions. At present, there is still no mature method that can solve both kinds of bias simultaneity. In this paper, our theoretical analysis shows that under certain assumptions, treatment effects can be unbiasedly estimated from observational data even in the presence of both confounding and collider biases. We propose the Selection Controlled CounterFactual Regression (SC-CFR) to simultaneously address both biases for treatment effect estimation. In SC-CFR, we first calculate the magnitude of the collider bias of different instances by estimating the selection model and then add a control term to remove the collider bias while learning a balanced representation to remove the confounding bias when estimating the outcome model. We conduct experiments on both synthetic and real-world datasets, and the results demonstrate that our method outperforms other baselines.

The main contributions in this paper are as follows: (1) We propose and investigate a practical problem on treatment effect estimation from observational data with both confounding and collider biases, which is still an open problem in causal inference to the best of our knowledge. (2) We propose a novel SC-CFR algorithm to estimate average treatment effect in observational studies with both confounding bias and collider bias. (3) Our theoretical analysis shows that both collider and confounding biases can be simultaneously removed under certain assumptions. (4) Extensive experiments show our proposed SC-CFR algorithm achieves a better performance of treatment effect estimation in observational studies with both synthetic and real-world datasets.

## 2 RELATED WORK

There are currently three main categories of treatment effect estimation methods in observational studies, i.e., methods based on the propensity score, confounder balancing and representation learning. The propensity score was introduced by (Rosenbaum & Rubin, 1983) and defined as $P(T = 1 \mid \mathbf{X})$. Based on the propensity score, various estimators have been proposed for treatment effect estimation with confounding bias, such as propensity score matching (Dehejia & Wahba, 2002) and inverse propensity weighting (IPW) (Hirano et al., 2003). Hirano & Imbens (2004) proposed the generalized propensity score for continuous treatments. Bang & Robins (2005) proposed the doubly robust estimator that combines IPW with regression. Confounder balancing is to learn sample weights that make the confounder distributions of control and treated units similar through sample re-weighting. Entropy Balancing (Hainmueller, 2012) was proposed to directly adjust sample weights to the specified sample moments while moving the sample weights as little as possible. Approximate Residual Balancing (Athey et al., 2018) combines balancing weights with a regularized regression adjustment through a lasso residual regression adjustment. Kuang et al. (2017) proposed Differentiated Confounder Balancing to select and differentiate confounders to balance the distributions. (Fong et al., 2018) proposed Covariate Balancing Generalized Propensity Score, which improves the estimation of propensity scores through combining confounder balancing techniques with the generalized propensity score. Methods based on deep representation learning were proposed to learn the balanced representation for all covariates, so that conditioning on the learned representation, the treatment is independent of the confounders. Johansson et al. (2016) proposed Balancing Neural Network to learn balanced representations through deep neural networks. Shalit et al. (2017) proposed Counterfactual Regression that applies integral probability metric to measure the distances between distributions. Yao et al. (2018) proposed a local similarity preserved individual treatment effect estimation method that preserves local similarity and balances data distributions simultaneously. (Hassanpour & Greiner, 2020) proposed Disentangled Representations for CounterFactual Regression that learns disentangled representations of confounders, adjustment variables and instrumental variables. All the above causal inference methods focused on solving confounding bias, while ignoring collider bias in observational studies.

In the literature of economics, there is some work focusing on correcting for sample selection bias. Heckman (1979) proposed a two-stage regression method called Heckman's Correction, and many extensions (Ogundimu & Hutton, 2016; Marchenko & Genton, 2012; Ding, 2014; Chib et al., 2009; Wiemann et al., 2022) have been proposed after that. These methods can only solve the simpler case of selection bias caused by covariates or the treatment variable, i.e., the missing at random (MAR) scenario, and suffer from confounding bias in data unless the model specification is correct. Many researchers regard sample selection bias as a special missing data problem, i.e., missing outcome, and discuss the effectiveness of causal inference methods to solve it (Smith & Elkan, 2004a; Daniel et al., 2012; Williamson et al., 2012), e.g., propensity score and doubly robust methods. Unfortunately, for the cases of missing not at random (MNAR), like collider bias, it remains an unidentifiable problem (Hernán & Robins, 2020). In the field of computer science, Bareinboim & Pearl (2012) were the first to classify and discuss the problem of causal inference under both selection bias and confounding bias, showing that with the help of some exogenous variables that satisfy certain conditions, it is feasible to solve collider bias and confounding bias simultaneously. Bareinboim et al. (2014); Bareinboim & Tian (2015) proposed a method to address selection bias by adjusting the selection backdoor path, and Correa & Bareinboim (2017) proposed a generalized adjustment method based on it to solve both confounding bias and selection bias. However, the above methods assume that the variables that determine the sample selection must satisfy certain conditions, e.g., $Y \perp\!\!\!\perp S \mid \mathbf{X}, T$, and thereby cannot solve the problem of collider bias.

## 3 SELECTION CONTROLLED COUNTERFACTUAL REGRESSION

### 3.1 PROBLEM FORMULATION

Suppose we have i.i.d. observational data $\mathcal{D} = \left\{ X_i, T_i, Y_i^{obs} \right\}_{i=1}^n$, where $n$ denotes the number of units. For the $i^{th}$ unit, we observe its treatment variable $T_i$, observed outcome variable $Y_i^{obs}$ and pre-treatment variables $X_i \in \mathbb{R}^{d \times 1}$, where $d$ denotes the dimension of the observed pre-treatment variables.

In this paper, we focus on the case of binary treatment, i.e., $T_i \in \{0, 1\}$, where $T_i = 1$ denotes unit $i$ is treated and $T_i = 0$ denotes otherwise. Under the potential outcome framework (Imbens & Rubin, 2015), we define the potential outcomes under treatment as $Y(1)$ and under control as $Y(0)$. Then the observed outcome $Y_i^{obs}$ can be written as:

$$Y_i^{obs} = \begin{cases} T_i \cdot Y_i(1) + (1 - T_i) \cdot Y_i(0) & S_i = 1 \\ NaN & S_i = 0 \end{cases}, \tag{1}$$

where $S_i$ indicates whether a unit indexed by $i$ is selected into the sample, and we cannot collect or observe the outcome of units with $S = 0$ (denoted by $NaN$). With the observational data, our goal is to estimate the Average Treatment effect (ATE), which is defined as:

$$ATE = E[Y(1) - Y(0)]. \tag{2}$$

However, for a selected unit with the treatment $T_i$ in dataset $\mathcal{D}$, we can only observe the outcome $Y_i(T_i)$ and the counterfactual outcome $Y_i(1 - T_i)$ is missing. What's worse, according to the definition of collider bias we mentioned earlier, $E[Y \mid T = 1, S = 1] \neq E[Y \mid T = 1]$. As a result, we cannot estimate ATE by Equation 2 directly.

To address this problem, we propose a novel Selection Controlled CounterFactual Regression (SC-CFR) to estimate ATE from observational data with both collider and confounding biases. And throughout this paper, we assume the **Stable Unit Treatment Value** assumption, the **overlap** assumption and the **Unconfoundedness** assumption (Imbens & Rubin, 2015) are satisfied.

## 3.2 PRELIMINARIES

In the presence of both confounding bias and collider bias, the general relationship among $\mathbf{X}$, $T$ and $Y$ can be represented by the additive noise model as:

$$Y = f(\mathbf{X}, T) + \epsilon_y. \tag{3}$$

And the mechanism that determines whether a unit is selected into the sample can be represented as:

$$S^* = h(\mathbf{X}, Y, T) + \epsilon_s, \tag{4}$$

where $S^*$ determines $S$ through $S = \begin{cases} 1 & S^* >= 0 \\ 0 & S^* < 0 \end{cases}$.

If we directly apply existing causal inference methods to reweight the samples with $S = 1$ to estimate the ATE, e.g., IPW or other confounder balancing methods, we can only address the confounding bias from $\mathbf{X}$ and ensure that $\mathbf{X}$ and $T$ of the reweighted sample are approximately independent, i.e., $P_w(\mathbf{X}, T, Y, S = 1) = P_w(\mathbf{X}) \cdot P_w(T) \cdot P_w(Y \mid \mathbf{X}, T, S = 1)$, but cannot address the collider bias from $S$, hence cannot make $P_w(\mathbf{X}, T, Y, S = 1) = P(\mathbf{X}, T, Y)$, leading to biased estimation.

And when we only use samples with $S = 1$ to perform Maximum Likelihood Estimation (MLE) of the outcome model to estimate counterfactuals and then the ATE, we will get the biased result $E[Y|T, \mathbf{X}, S = 1] = f(\mathbf{X}, T) + E[\epsilon_y \mid S = 1]$, where $E[\epsilon_y \mid S = 1]$ is **the biased term**. Therefore, if we can estimate $E[\epsilon_y \mid S = 1]$ correctly and add it to the regression as a control variable, it will be possible to achieve an unbiased estimate of $f(\mathbf{X}, T)$. We show that under the following assumption, we can estimate $E[\epsilon_y \mid S = 1]$ using only observational data, even in the presence of both confounding bias and collider bias.

**Assumption 1.** The noise term $\epsilon_y$ and $\epsilon_s$ are both additive. And $\epsilon_y$ is additive and remains the same distribution type after being transformed by the selection mechanism function, i.e., $h(\mathbf{X}, f(\mathbf{X}, T) + \epsilon_y, T) = h(\mathbf{X}, f(\mathbf{X}, T), T) + \delta(\epsilon_y)$, where $\delta(\epsilon_y)$ denotes the transformed term by the selection mechanism function.

**Proposition 1.** Under Assumption 1, the biased term $E[\epsilon_y \mid S = 1]$ caused by the collider bias can be converted into the conditional expectation about the noise terms.

**Proof.** Under Assumption 1, we have

$$\begin{aligned} E[\epsilon_y \mid S = 1] &= E[\epsilon_y \mid S^* >= 0] \\ &= E[\epsilon_y \mid h(\mathbf{X}, Y, T) + \epsilon_s >= 0] \\ &= E[\epsilon_y \mid h(\mathbf{X}, f(\mathbf{X}, T) + \epsilon_y, T) + \epsilon_s >= 0] \\ &= E[\epsilon_y \mid h(\mathbf{X}, f(\mathbf{X}, T), T) + \delta(\epsilon_y) + \epsilon_s >= 0] \\ &= E[\epsilon_y \mid \delta(\epsilon_y) + \epsilon_s >= -h(\mathbf{X}, f(\mathbf{X}, T), T)], \end{aligned} \tag{5}$$

Through Equation 5, we can convert the biased term into the conditional expectation about the noise terms, which is computable if we make certain assumptions about the distribution of $\epsilon_y$ and $\epsilon_s$. Considering that the most widely used loss function in machine learning is the Mean Square Error (MSE) function, which assumes that the noise term satisfies the Gaussian distribution, to avoid conflicts among assumptions, we make the following assumption about the noise term.

**Assumption 2.** $\epsilon_y$ and $\epsilon_s$ satisfy that $\epsilon_y \sim N(0, \sigma_y^2)$, $\epsilon_s \sim N(0, \sigma_s^2)$ and $\begin{pmatrix} \epsilon_y \\ \epsilon_s \end{pmatrix} \sim N\left( \begin{pmatrix} 0 \\ 0 \end{pmatrix}, \begin{pmatrix} \sigma_y^2 & \rho \\ \rho & \sigma_s^2 \end{pmatrix} \right)$.

**Proposition 2.** Under assumption 2, the biased term $E[\epsilon_y \mid \delta(\epsilon_y) + \epsilon_s >= -h(\mathbf{X}, f(\mathbf{X}, T), T)]$ caused by the collider bias can be calculated and controlled while solving the confounding bias by representation learning.

**Proof.** Based on the theorem of moments of the incidentally truncated bivariate Normal distribution (Greene, 2017), we have

$$
\begin{aligned}
E[\epsilon_y \mid \delta(\epsilon_y) + \epsilon_s >= -h(\mathbf{X}, f(\mathbf{X}, T), T)] &= \frac{cov(\epsilon_y, \delta(\epsilon_y) + \epsilon_s)}{std(\delta(\epsilon_y) + \epsilon_s)} \cdot \frac{\phi(-h(\mathbf{X}, f(\mathbf{X}, T), T)/std(\delta(\epsilon_y) + \epsilon_s))}{\Phi(h(\mathbf{X}, f(\mathbf{X}, T), T)/std(\delta(\epsilon_y) + \epsilon_s)))} \\
&= \frac{E[\epsilon_y \cdot \delta(\epsilon_y)] + E[\epsilon_y \epsilon_s]}{std(\delta(\epsilon_y) + \epsilon_s)} \cdot \frac{\phi(-h(\mathbf{X}, f(\mathbf{X}, T), T)/std(\delta(\epsilon_y) + \epsilon_s))}{\Phi(h(\mathbf{X}, f(\mathbf{X}, T), T)/std(\delta(\epsilon_y) + \epsilon_s)))},
\end{aligned}
\tag{6}
$$

where $cov(\cdot)$ denotes the covariance, $std(\cdot)$ denotes the standard deviation, $\phi(\cdot)$ and $\Phi(\cdot)$ denote the density and distribution function for a standard normal variable, respectively.

Let $\alpha = \frac{1}{std(\delta(\epsilon_y) + \epsilon_s)}$, $\lambda(\alpha \cdot h(\mathbf{X}, f(\mathbf{X}, T), T)) = \frac{\phi(-\alpha \cdot h(\mathbf{X}, f(\mathbf{X}, T), T))}{\Phi(\alpha \cdot h(\mathbf{X}, f(\mathbf{X}, T), T))}$ and $\beta = \frac{E[\epsilon_y \cdot \delta(\epsilon_y)] + E[\epsilon_y \epsilon_s]}{std(\delta(\epsilon_y) + \epsilon_s)}$, through Equations 5 and 6, we have

$$
E[Y \mid \mathbf{X}, T, S = 1] = f(\mathbf{X}, T) + \beta \cdot \lambda(\alpha \cdot h(\mathbf{X}, f(\mathbf{X}, T), T)).
\tag{7}
$$

Since $\alpha$ and $\beta$ are constants, we can consider them as parameters of regression, and thus we can calculate and control it by firstly estimating $h(\mathbf{X}, f(\mathbf{X}, T), T)$ and learning $\alpha$, and then adding $\lambda(\alpha \cdot h(\mathbf{X}, f(\mathbf{X}, T), T))$ as a control term in the regression. At the same time, in order to avoid confounding bias, we still need to eliminate the influence of confounders on the treatment variable during regression, which can be achieved by learning representations of covariates, denoted by $R(\mathbf{X})$, to make $P(R(\mathbf{X}), Y, T = 1, S = 1) = P(R(\mathbf{X}), Y, T = 0, S = 1)$ and $g(R(\mathbf{X}), T)) = f(\mathbf{X}, T)$ (Johansson et al., 2016). Then we can estimate the ATE by

$$
\widehat{ATE} = \frac{1}{n_s} \sum_{i:S_i=1} (\widehat{g}(R(\mathbf{X_i}), 1) - \widehat{g}(R(\mathbf{X_i}), 0)),
\tag{8}
$$

where $n_s$ denotes the number of selected units.

Based on the above assumptions and propositions, we propose a method to solve both collider bias and confounding bias simultaneously.

### 3.3    ALGORITHM AND OPTIMIZATION

With the above preliminaries, we propose a novel method, named Selection Controlled CounterFactual Regression (SC-CFR), to estimate the ATE on observational data with confounding bias and collider bias. We implement our method through deep neural networks, as shown in Figure 2.

According to the analyses in Section 3.2, we have the following objectives in our training process in order to solve collider bias and confounding bias simultaneously. First, we need to use the treatment variable and covariates of samples with both $S = 0$ and $S = 1$ to regress the selection mechanism model and obtain an accurate estimate of $h(\mathbf{X}, f(\mathbf{X}, T), T)$. Second, we need to learn the parameter $\alpha$ and calculate $\lambda(\alpha \cdot h(\mathbf{X}, f(\mathbf{X}, T), T))$. Third, we need to learn representations $R(\mathbf{X})$ to solve the confounding bias. And fourth, we need to add the control term $\lambda(\alpha \cdot h(\mathbf{X}, f(\mathbf{X}, T), T))$ to the outcome regression and learn the parameters in $g(R(\mathbf{X_i}), T)$ and $\beta$.

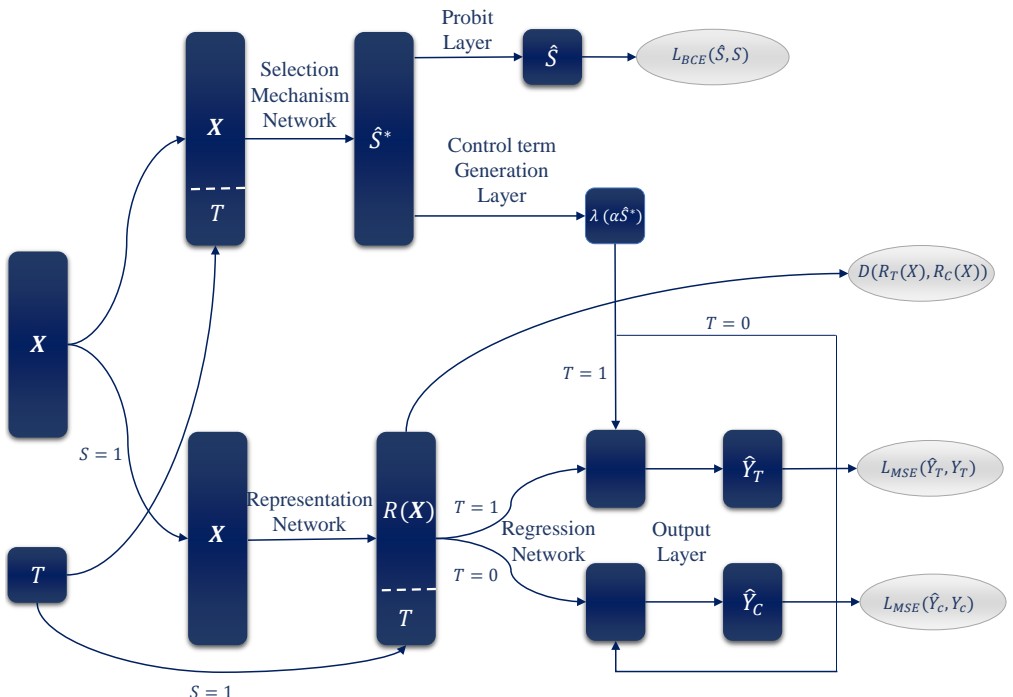

Figure 2: Neural network architecture of the proposed SC-CFR algorithm.

To achieve the above goals, we define our objective function as

$$
\begin{aligned}
\min \quad & \frac{1}{n} \sum_{i=1}^{n} L_{BCE}\left(h\left(\mathbf{X_i}, f\left(\mathbf{X_i}, T_i\right), T_i\right)\right), S_i\right) \\
& + \frac{1}{n_s} \sum_{i:S_i=1} w_i \cdot L_{MSE}\left(g\left(R\left(\mathbf{X_i}\right), T_i\right) + \beta \cdot \lambda(\alpha \cdot h\left(\mathbf{X_i}, f\left(\mathbf{X_i}, T_i\right), T_i\right)), Y_i^{obs}\right) \\
& + \gamma \cdot D\left(R\left(\mathbf{X_i}\right)_{i:T_i=0, S_i=1}, R\left(\mathbf{X_i}\right)_{i:T_i=1, S_i=1}\right) \\
& + \Omega(h, g, \alpha, \beta),
\end{aligned} \tag{9}
$$

where $L_{BCE}$ denotes the Binary Cross Entropy (BCE) loss function, $L_{MSE}$ denotes the MSE loss function, $\Omega(\cdot)$ denotes the regularization terms, $w_i$ denotes the compensation weights for the difference in treatment group size and $D(\cdot)$ denotes the metric that measures the distance between distributions, which can be the Integral Probability Metric (IPM) (Shalit et al., 2017). We also use batch normalization (Ioffe & Szegedy, 2015), early stopping and other optimization methods in the training process. For details of our SC-CFR algorithm, please see Figure 2.

## 4 EXPERIMENTS

### 4.1 BASELINES

We implement the following baseline estimators **Direct Estimator** (Dir) that estimates ATE through calculating $E\left[Y^{obs} \mid S=1\right] - E\left[Y^{obs} \mid S=1\right]$ directly, **Inverse Probability Weighting** (IPW) (Hirano et al., 2003), **Heckman's Correction** (Heckit) (Heckman, 1979), **Differentiated Confounder Balancing** (DCB) (Kuang et al., 2017) and **Counterfactual Regression** (CFR) (Shalit et al., 2017), which are respectively very representative in each category of causal inference methods, to estimate the ATE and compare them with our proposed estimator (SC-CFR). Note that we use the Maximum Mean Discrepancy (MMD) metric to implement CFR and SC-CFR, and we split each dataset into 60/20/20 train/validation/test datasets. We implement the algorithm in PyTorch environment with Python 3.8. The CPU we use is Intel(R) Core(TM) i7-8750H CPU @ 2.20 GHz and the GPU we use is NVIDIA GeForce GTX 1050 Ti with CUDA version 10.2.

Table 1: Results on synthetic datasets under different confounding bias strengths $s_c$ with a fixed collider bias strength $s_s = 3$. The smaller $Bias$, $SD$, $MAE$ and $RMSE$, the better.

|           | $s_c = 10$ | | | | $s_c = 50$ | | | | $s_c = 100$ | | | |
|-----------|------|------|------|------|------|------|------|------|------|------|------|------|
| Estimator | $Bias$ | $SD$ | $MAE$ | $RMSE$ | $Bias$ | $SD$ | $MAE$ | $RMSE$ | $Bias$ | $SD$ | $MAE$ | $RMSE$ |
| Dir       | 1.275 | 0.095 | 1.275 | 1.279 | 1.151 | 0.244 | 1.151 | 1.177 | 0.960 | 0.514 | 0.998 | 1.089 |
| IPW       | 5.283 | 0.649 | 5.283 | 5.323 | 9.357 | 3.889 | 9.357 | 10.133 | 12.726 | 13.480 | 12.726 | 18.538 |
| Heckit    | 0.756 | **0.075** | 0.756 | 0.760 | 4.296 | 0.403 | 4.296 | 4.315 | 7.415 | 1.269 | 7.415 | 7.523 |
| DCB       | 0.979 | 0.084 | 0.979 | 0.983 | 2.511 | **0.212** | 2.511 | 2.520 | 3.020 | **0.510** | 3.020 | 3.063 |
| CFR       | 1.473 | 0.241 | 1.473 | 1.493 | 0.885 | 0.349 | 0.885 | 0.951 | 1.017 | 0.616 | 1.048 | 1.189 |
| SC-CFR    | **0.575** | 0.317 | **0.599** | **0.657** | **0.401** | 0.404 | **0.460** | **0.569** | **0.568** | 0.765 | **0.794** | **0.953** |

Based on the estimated ATE, we calculate its Bias, standard deviations (SD), mean absolute errors (MAE) and root mean square errors (RMSE) to evaluate the performance of the above estimators.

### 4.2 EXPERIMENTS ON SYNTHETIC DATA

#### 4.2.1 DATASETS

In order to better evaluate the performance of each estimator in the presence of both confounding bias and collider bias, we generate synthetic datasets with different strengths of collider bias and confounding bias, denoted by $s_s$ and $s_c$ respectively, where $s_s$ affects the difference between the population distribution and the sample distribution (i.e., denotes the strength of collider bias), and $s_c$ affects the number of confounders (i.e., denotes the strength of confounding bias). The size $n$ of our generated datasets is 10,000.

We first generate the continuous pre-treatment variables $\mathbf{X} \in \mathbb{R}^{n \times s_c}$ with independent Gaussian distributions as $\mathbf{X} \overset{\text{i.i.d.}}{\sim} N(\mathbf{0}, \mathbf{1})$. To introduce confounding bias with strength $s_c$ into datasets, we generate the binary treatment variable $T \in \mathbb{R}^{n \times 1}$ from a logistic function as $T \sim Bernoulli\left(1/\left(1 + e^{-\sum_{i=1}^{s_c}(\mathbf{1}(mod(i,2) \equiv 1) \cdot X_i/2 - \mathbf{1}(mod(i,2) \neq 1) \cdot X_i/2 + \epsilon_t)}\right)\right)$, where $Bernoulli(\cdot)$ denotes the Bernoulli distribution, $\mathbf{1}(\cdot)$ is the indicator function, function $mod(x, y)$ returns the modulus after division of $x$ by $y$ and $\epsilon_t \sim N(0, 1)$. Next, we generate the continuous outcome variable $Y \in \mathbb{R}^{n \times 1}$ from a non-linear function as $Y = 3 \cdot T + \sum_{i=1}^{s_c}\left(T \cdot Z_i + (\mathbf{1}(mod(i, 2) \neq 1) - \mathbf{1}(mod(i, 2) \equiv 1)) \cdot \left(\frac{mod(i,2)+1}{2}\right) \cdot \left(Z_i + Z_i^2\right)\right) + \epsilon_y$, where $\epsilon_y \sim N(0, 1)$. To introduce collider bias with strength $s_s$ into datasets, we generate the binary selection variable $S \in \mathbb{R}^{n \times 1}$ from a logistic function as $S \sim Bernoulli\left(1/\left(1 + e^{-s_s \cdot (Y - T + \epsilon_s)}\right)\right)$, where $\epsilon_s \sim N(0, 1)$ and a unit is selected into the sample only when $S = 1$. The ground truth ATE can be calculated easily by the above functions.

To compare our estimator with baselines under different strengths of confounding bias and collider bias, we first fix $s_c$ to 50 and evaluate the performance of each estimator under $s_s = \{1, 3, 5\}$, then fix $s_s$ to 3 and evaluate the performance of each estimator under $s_c = \{10, 50, 100\}$. We independently performed 50 experiments under each setting and regenerated the dataset for each experiment to evaluate the robustness of our estimator.

#### 4.2.2 RESULTS

We report the results in Table 1 for comparing among different confounding bias strengths $s_c$ with the collider bias strength $s_s$ fixed and Table 2 for comparing among different $s_s$ with $s_c$ fixed.

From Table 1, our observations and interpretations are as follows: In general, the performance of all estimators gradually decreases as the strength of confounding bias increases. The overall performance of IPW is very poor, because IPW does reduce confounding bias in data, but in the setting of high dimensional variables, it may be more likely to suffer from incorrect model specification and extreme propensity scores. Heckman's Correction performs better than the direct estimator only when $s_c = 10$, because it only focuses on the collider bias and may suffer more from confounding bias in high dimensional settings due to model misspecification, as we mentioned earlier. The performance of DCB is much better than Heckman's Correction, since it directly estimates ATE from the re-weighted observational data. However, it ignores collider bias during the learning process of sample weights, which makes the distribution of re-weighted samples shift more from the overall distribution, leading to biased estimation. CFR estimator has better performance than the above

Table 2: Results on synthetic datasets under different collider bias strengths $s_s$ with a fixed confounding bias strength $s_c = 50$. The smaller $Bias$, $SD$, $MAE$ and $RMSE$, the better.

| Estimator | $s_s = 1$ | | | | $s_s = 3$ | | | | $s_s = 5$ | | | |
|---|---|---|---|---|---|---|---|---|---|---|---|---|
| | $Bias$ | $SD$ | $MAE$ | $RMSE$ | $Bias$ | $SD$ | $MAE$ | $RMSE$ | $Bias$ | $SD$ | $MAE$ | $RMSE$ |
| Dir | 1.177 | 0.248 | 1.177 | 1.203 | 1.151 | 0.244 | 1.151 | 1.177 | 1.146 | 0.264 | 1.146 | 1.176 |
| IPW | 9.976 | 4.359 | 9.976 | 10.887 | 9.357 | 3.889 | 9.357 | 10.133 | 12.154 | 6.248 | 12.154 | 13.666 |
| Heckit | 4.249 | 0.356 | 4.249 | 4.264 | 4.296 | 0.403 | 4.296 | 4.315 | 4.383 | 0.399 | 4.383 | 4.402 |
| DCB | 2.255 | **0.221** | 2.255 | 2.266 | 2.511 | **0.212** | 2.511 | 2.520 | 2.441 | **0.231** | 2.441 | 2.452 |
| CFR | 0.917 | 0.407 | 0.917 | 1.003 | 0.885 | 0.349 | 0.885 | 0.951 | 1.007 | 0.417 | 1.007 | 1.090 |
| SC-CFR | **0.219** | 0.482 | **0.411** | **0.529** | **0.401** | 0.404 | **0.460** | **0.569** | **0.611** | 0.382 | **0.641** | **0.721** |

estimators, but it is easier to overfit on the selected units, resulting in large counterfactual prediction errors, and thus more susceptible to collider bias. The direct estimator is overall the most stable one among the baselines since other methods for a specific bias are more vulnerable to the harmful impact of the other bias, but it still suffers from both biases. Our SC-CFR effectively solves both confounding bias and collider bias in observational data and performs best in almost all cases. Note that the reason for the relatively mediocre SD performance of SC-CFR is that in our implementation, there may be too little data with $S = 1$ compared to the data with $S = 0$ in each batch, resulting in slow fitting when the batch size is relatively small (but still get better performance than all the baselines).

From Table 2, our observations and interpretations are as follows: The performance of IPW is overall the worst because it suffers from incorrect model specification as we analyzed earlier. The Heckman's Correction estimator performs poorly in all cases for the same reason we mentioned earlier. DCB performs slightly better than the Heckman's Correction estimator since it is a re-weighting method and can tolerate a certain degree of incorrect model specification, but still suffers from collider bias. The direct estimator is overall stable, although it is helpless against both biases. CFR achieves better performance than other baselines, but is susceptible to collider bias as we mentioned before. Our SC-CFR performs best in most of time and it proves that our method does solve both collider and confounding biases in observational studies.

## 4.3 EXPERIMENTS ON REAL-WORLD DATA

### 4.3.1 DATASETS

In order to evaluate the proposed method in real-world scenarios, we conduct experiments on two well-known datasets: the IHDP dataset (Hill, 2011) and the Lalonde dataset (LaLonde, 1986).

**The IHDP dataset:** The original RCT data of the Infant Health and Development Program (IHDP) aims at evaluating the effect of specialist home visits on the future cognitive test scores of premature infants (Brooksgunn et al., 1992). In (Hill, 2011), they removed a non-random subset of the treated group and used simulated outcomes to induce confounding bias. As (Shalit et al., 2017) did, we use the simulated outcome implemented as setting "A" in the NPCI package (Dorie, 2016).[1] To introduce collider bias into the IHDP dataset, we generate the binary selection variable $S$ from a logistic function as $S \sim Bernoulli\left(1/\left(1 + e^{-(5 \cdot T - (1/2) \cdot Y + \epsilon_{ihdp})}\right)\right)$, where $\epsilon_{ihdp} \sim N(0, 1)$. Intuitively, parents whose child had lower cognitive abilities and received home visits were more likely to report the test scores, leading to collider bias. The final dataset comprises 747 units (139 treated, 608 control) with 26 pre-treatment variables related to the children and their families.

**The Lalonde dataset:** It is a widely used benchmark in the causal inference community based on the RCT, aiming to estimate the effect of job training programs on future incomes. It combines a randomized study based on the National Supported Work (NSW) program with observational data to form a larger dataset (LaLonde, 1986). Guided by (Hainmueller, 2012; A. Smith & E. Todd, 2005), to introduce confounding bias, we use the Dehejia and Wahha sampled dataset of the LaLonde (185 treated, 260 control) (Dehejia & Wahba, 2002), and replace its control group with the control group (2490 control) from Population Survey of Income Dynamics (PSID) where the measured covariates are the same with the experimental data.[2] To introduce collider bias into the Lalonde dataset, we generate the binary selection variable $S$ from a logistic function as

---

[1]The dataset is available at http://www.fredjo.com/

[2]The datasets are available at https://users.nber.org/ rdehejia/nswdata2.html

Table 3: Results on real datasets. The smaller $Bias$, $SD$, $MAE$ and $RMSE$, the better.

| Estimator | IHDP | | | | Lalonde | | | |
|---|---|---|---|---|---|---|---|---|
| | $Bias$ | $SD$ | $MAE$ | $RMSE$ | $Bias$ | $SD$ | $MAE$ | $RMSE$ |
| Dir | 0.520 | **0.092** | 0.520 | 0.529 | 6.673 | **0.195** | 6.676 | 3.577 |
| IPW | 26.388 | 1.006 | 26.388 | 26.407 | 67.512 | 39.771 | 67.512 | 78.355 |
| Heckit | 1.381 | 1.159 | 1.507 | 1.803 | 3.577 | 1.247 | 3.577 | 3.788 |
| DCB | 0.359 | 0.094 | 0.359 | 0.371 | 1.392 | 0.295 | 1.392 | 1.422 |
| CFR | 0.508 | 0.176 | 0.508 | 0.538 | 1.781 | 0.822 | 1.799 | 1.961 |
| SC-CFR | **0.049** | 0.264 | **0.215** | **0.306** | **0.286** | 0.804 | **0.667** | **0.853** |

$S \sim Bernoulli \left( 1 / \left( 1 + e^{-(Y - 3 \cdot T + \epsilon_{lalonde})} \right) \right)$, where $\epsilon_{lalonde} \sim N(0, 1)$. Intuitively, ones who had a lower income and not participated in job training programs were unwilling to report their incomes, leading to collider bias. The final dataset comprises 2675 units (185 treated, 2490 control) with 11 pre-treatment variables related to individuals' basic information, ethnicity and job information.

### 4.3.2 RESULTS

We report the results in Table 3 and our findings are as follows: The overall performance on the IHDP dataset is much better since the outcomes of the IHDP dataset are simulated with relatively simple functions. Specifically, IPW performs very poorly on both datasets, because the propensity score model is more complex on the real dataset, leading to serious errors in propensity score estimation. Heckman's Correction performs much better than the IPW estimator, but still suffer from model misspecification. The direct estimator achieves very good performance on the IHDP dataset, but performs poorly on the Lalonde dataset since the Lalonde dataset is completely collected in real-world scenarios and thus has stronger confounding bias. It also suffers from collider bias. On both datasets, CFR is better than the above estimators because it solves confounding bias in data and has stronger fitting ability, but is very susceptible to collider bias as we analyzed before. DCB is the one with the best overall performance among all baselines, but still suffer from collider bias. SC-CFR we propose performs best in most of time on both datasets. It proves that our method can effectively solve both confounding bias and collider bias in real-world scenarios and achieve a more precise treatment effect estimation.

## 5 CONCLUSION AND FUTURE WORK

In this paper, we focus on the problem of estimating treatment effect in observational studies with both confounding bias and collider bias. We argue that previous methods mainly focus on solving either confounding bias or selection bias caused by only the treatment variable or covariates, while ignoring collider bias in data, and thus underperform in the presence of both biases. Therefore, we propose an algorithm, named the Selection Controlled CounterFactual Regression, to simultaneously address both biases for treatment effect estimation. We first calculate the magnitude of the collider bias of different instances by estimating the selection model and then add a control term to remove the collider bias while learning a balanced representation to remove the confounding bias when estimating the outcome model. And the experiment results on synthetic datasets and real-world datasets demonstrate that our estimator outperforms other baselines.

One of the limitations of our work is that the proposed model is integrated, and when the size of samples with $S = 1$ is much smaller than that with $S = 0$, it is likely to have inadequate $T = 1$ or $T = 0$ data for model training in each batch, resulting in slow fitting. A simple solution is to divide the integrated model into independent models, i.e., to train the selection mechanism model and the outcome model step by step, to avoid the above problem. At the same time, our algorithm is based on the unconfoundedness assumption and thus cannot solve the confounding bias caused by unobserved confounding variables. Therefore, our future work will focus on causal inference with unobserved confounders in observational studies with both confounding bias and collider bias, and developing more general methods to estimate the biased control term, which can relax the noise distribution assumptions.

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

# A APPENDIX

## A.1 SUPPLEMENTARY EXPERIMENTS WHEN ASSUMPTIONS ARE NOT SATISFIED

Our approach is based on the following assumptions.

**Assumption 1.** The noise term $\epsilon_y$ and $\epsilon_s$ are both additive. And $\epsilon_y$ is additive and remains the same distribution type after being transformed by the selection mechanism function, i.e., $h(\mathbf{X}, f(\mathbf{X}, T) + \epsilon_y, T) = h(\mathbf{X}, f(\mathbf{X}, T), T) + \delta(\epsilon_y)$, where $\delta(\epsilon_y)$ denotes the transformed term by the selection mechanism function.

**Assumption 2.** $\epsilon_y$ and $\epsilon_s$ satisfy that $\epsilon_y \sim N(0, \sigma_y^2)$, $\epsilon_s \sim N(0, \sigma_s^2)$ and $\begin{pmatrix} \epsilon_y \\ \epsilon_s \end{pmatrix} \sim N\left(\begin{pmatrix} 0 \\ 0 \end{pmatrix}, \begin{pmatrix} \sigma_y^2 & \rho \\ \rho & \sigma_s^2 \end{pmatrix}\right)$.

To verify the robustness of our method when the above two assumptions are not satisfied, we conduct the following supplementary experiments on synthetic data.

First, we change the noise terms from Gaussian distribution to uniform distribution that violates Assumption 2 and report the results in Table 4 and Table 5. We also change the selection mechanism function with $\epsilon_y \cdot T$ that violates Assumption 1 and report the results in Table 6 and Table 7. The results show that even if the assumptions are not satisfied, the performance of SC-CFR only decreases slightly and still is the best among all methods.

Table 4: Results on synthetic datasets that violate Assumption 2 under different confounding bias strengths $s_c$ with a fixed collider bias strength $s_s = 3$. The smaller $Bias$, $SD$, $MAE$ and $RMSE$, the better.

| Estimator | $s_c = 10$ | | | | $s_c = 50$ | | | | $s_c = 100$ | | | |
|---|---|---|---|---|---|---|---|---|---|---|---|---|
| | $Bias$ | $SD$ | $MAE$ | $RMSE$ | $Bias$ | $SD$ | $MAE$ | $RMSE$ | $Bias$ | $SD$ | $MAE$ | $RMSE$ |
| Dir | 1.532 | 0.084 | 1.874 | 1.876 | 1.319 | 0.233 | 1.319 | 1.340 | 1.546 | **0.108** | 1.546 | 1.549 |
| IPW | 4.650 | 0.700 | 4.650 | 4.703 | 9.862 | 5.338 | 9.862 | 11.214 | 10.435 | 13.119 | 10.435 | 16.763 |
| Heckit | 1.518 | 0.140 | 1.518 | 1.524 | 3.456 | 0.352 | 3.456 | 3.474 | 6.719 | 1.147 | 6.719 | 6.816 |
| DCB | 1.870 | **0.081** | 1.870 | 1.872 | 2.933 | **0.225** | 2.933 | 2.942 | 3.383 | 0.491 | 3.383 | 3.419 |
| CFR | 1.634 | 0.312 | 1.634 | 1.663 | 0.935 | 0.361 | 0.935 | 1.002 | 0.990 | 0.563 | 1.010 | 1.140 |
| SC-CFR | **0.699** | 0.416 | **0.714** | **0.813** | **0.652** | 0.573 | **0.779** | **0.868** | **0.824** | 0.397 | **0.836** | **0.915** |

Table 5: Results on synthetic datasets that violate Assumption 2 under different collider bias strengths $s_s$ with a fixed confounding bias strength $s_c = 50$. The smaller $Bias$, $SD$, $MAE$ and $RMSE$, the better.

| Estimator | $s_s = 1$ | | | | $s_s = 3$ | | | | $s_s = 5$ | | | |
|---|---|---|---|---|---|---|---|---|---|---|---|---|
| | $Bias$ | $SD$ | $MAE$ | $RMSE$ | $Bias$ | $SD$ | $MAE$ | $RMSE$ | $Bias$ | $SD$ | $MAE$ | $RMSE$ |
| Dir | 1.343 | 0.277 | 1.343 | 1.371 | 1.319 | 0.233 | 1.319 | 1.340 | 1.286 | 0.231 | 1.286 | 1.306 |
| IPW | 9.091 | 8.648 | 9.113 | 12.548 | 9.862 | 5.338 | 9.862 | 11.214 | 13.372 | 11.166 | 13.372 | 17.421 |
| Heckit | 3.225 | 0.433 | 3.225 | 3.254 | 3.456 | 0.352 | 3.456 | 3.475 | 3.375 | 0.418 | 3.375 | 3.401 |
| DCB | 1.887 | **0.253** | 1.887 | 1.904 | 2.933 | **0.225** | 2.933 | 2.942 | 2.836 | **0.204** | 2.836 | 2.843 |
| CFR | 0.826 | 0.464 | 0.854 | 0.948 | 0.935 | 0.361 | 0.935 | 1.002 | 0.944 | 0.421 | 0.944 | 1.034 |
| SC-CFR | **0.142** | 0.454 | **0.401** | **0.476** | **0.652** | 0.573 | **0.779** | **0.868** | **0.644** | 0.447 | **0.667** | **0.783** |

Table 6: Results on synthetic datasets that violate Assumption 1 under different confounding bias strengths $s_c$ with a fixed collider bias strength $s_s = 3$. The smaller $Bias$, $SD$, $MAE$ and $RMSE$, the better.

| Estimator | $s_c = 10$ | | | | $s_c = 50$ | | | | $s_c = 100$ | | | |
|---|---|---|---|---|---|---|---|---|---|---|---|---|
| | $Bias$ | $SD$ | $MAE$ | $RMSE$ | $Bias$ | $SD$ | $MAE$ | $RMSE$ | $Bias$ | $SD$ | $MAE$ | $RMSE$ |
| Dir | 1.425 | 0.103 | 1.425 | 1.428 | 1.383 | 0.257 | 1.383 | 1.406 | 1.174 | **0.438** | 1.174 | 1.253 |
| IPW | 4.821 | 0.776 | 4.821 | 4.883 | 17.505 | 38.837 | 17.505 | 42.600 | 16.195 | 28.827 | 16.195 | 33.065 |
| Heckit | 0.967 | 0.510 | 0.967 | 1.092 | 4.149 | 0.354 | 4.149 | 4.164 | 7.284 | 1.199 | 7.284 | 7.382 |
| DCB | 0.854 | **0.071** | 0.854 | 0.857 | 2.322 | **0.234** | 2.322 | 2.334 | 2.865 | 0.458 | 2.865 | 2.901 |
| CFR | 1.710 | 0.232 | 1.710 | 1.725 | 1.031 | 0.403 | 1.031 | 1.107 | 1.038 | 0.598 | 1.080 | 1.198 |
| SC-CFR | **0.650** | 0.311 | **0.659** | **0.721** | **0.646** | 0.505 | **0.684** | **0.820** | **0.319** | 0.680 | **0.587** | **0.751** |

Table 7: Results on synthetic datasets that violate Assumption 1 under different collider bias strengths $s_s$ with a fixed confounding bias strength $s_c = 50$. The smaller $Bias$, $SD$, $MAE$ and $RMSE$, the better.

| Estimator | $s_s = 1$ | | | | $s_s = 3$ | | | | $s_s = 5$ | | | |
|---|---|---|---|---|---|---|---|---|---|---|---|---|
| | $Bias$ | $SD$ | $MAE$ | $RMSE$ | $Bias$ | $SD$ | $MAE$ | $RMSE$ | $Bias$ | $SD$ | $MAE$ | $RMSE$ |
| Dir | 1.159 | **0.206** | 1.159 | 1.177 | 1.383 | 0.257 | 1.383 | 1.406 | 1.346 | 0.238 | 1.346 | 1.367 |
| IPW | 10.231 | 7.256 | 10.231 | 12.543 | 17.505 | 38.837 | 17.505 | 42.600 | 12.430 | 13.998 | 12.430 | 18.720 |
| Heckit | 4.318 | 0.271 | 4.318 | 4.327 | 4.149 | 0.354 | 4.149 | 4.164 | 4.229 | 0.287 | 4.229 | 4.239 |
| DCB | 2.269 | 0.210 | 2.269 | 2.279 | 2.322 | **0.234** | 2.322 | 2.334 | 2.259 | **0.192** | 2.259 | 2.267 |
| CFR | 0.782 | 0.361 | 0.783 | 0.862 | 1.031 | 0.403 | 1.031 | 1.107 | 1.036 | 0.360 | 1.036 | 1.097 |
| SC-CFR | **0.233** | 0.520 | **0.421** | **0.570** | **0.646** | 0.505 | **0.684** | **0.820** | **0.747** | 0.384 | **0.747** | **0.840** |

