# OpenReview forum: "Treatment Effect Estimation with Collider Bias and Confounding Bias"
_ICLR.cc/2023/Conference — Submitted to ICLR 2023_

### Official Review · Reviewer_e76D · 2022-10-22

**Confidence:** 4
**Correctness:** 3
**Technical Novelty And Significance:** 2
**Empirical Novelty And Significance:** 2
**Recommendation:** 5

**Clarity, Quality, Novelty And Reproducibility:**

Clarity: Good. The paper is well organized, but the presentation can be improved further.
Quality: Need to revise the experiment part.
Novelty: moderate. The practical impact is low.
Reproducibility: Most details are described such that an expert should be able to reproduce the main results.


**Strength And Weaknesses:**

S1: The paper focuses on an interesting problem of simultaneously estimating unbiased ATE when there is confounder and collider bias in the observational data.

S2: This paper proposes the SC-CFR algorithm, which learns the magnitude of selection bias through the neural network, and then eliminates collider and confounding bias by adding control items and balance items.

W1: The solution does not have a strong practical implication. The key to collider bias removal is to estimate h(X, f(X, T), T). “We need to use the treatment variable and covariates of samples with both S = 0 and S = 1 to regress the selection mechanism model and obtain an accurate estimate of h (X, f (X, T) , T)).” Actually, once there is an accurate h(), the problem of collider bias is largely solved. The balance can be achieved by a neural network method as in this paper or other means. That is not crucial. The crucial point is that there are no unbiased samples with both S=1 and S=1 in most real world cases. Therefore, I am not so excited about the method.

W2: This paper focuses only on simple collider bias. There are a few other more complex ones (Hernán MA, Robins JM (2020). Causal Inference: What If. Boca Raton: Chapman & Hall/CRC.) In the real-world dataset, S is manually added by simulation.


W3: Other comments for clarifications.
a)	In 4.2.2, “We report the results in Table 1 for comparing among different confounding bias strengths sc with the collider bias strength ss fixed and Table 2 for comparing among different sc with ss fixed.”. Table 1 is the result when ss is fixed and Table 2 is the result when sc is fixed.
b)	In 4.2.2, “In general, the performance of all estimators gradually decreases as the strength of confounding bias increases”. This description confuses me because the direct estimator improves performance when the number of confounders is increased.
c)	In 4.2.2 Table 1, When sc = 50, CFR and SC-CFR achieve the best performance compared with the setting of other strengths of collider bias. This needs to be explained.


**Summary Of The Paper:**

This paper studies reducing selection bias and confounders bias in estimating average causal effects (ATE) in observational data. This paper provides the SC-CFR algorithm for simultaneously addressing confounder and collider bias to achieve unbiased ATE estimation. Specifically, SC-CFR first computes the magnitude of the collider bias, and then adds a control term to remove the collider bias, and then learns a balanced representation to remove the confounding bias. Extensive empirical results on synthetic and real-world datasets show that their method consistently outperforms benchmarks for treatment effect estimates when both types of bias are present.

**Summary Of The Review:**

It is a technically solid, modest impact paper, with no major concerns with respect to quality and reproducibility.

I am not excited about the paper since its practical impact is low.

---

> ### Author Response · Authors · 2022-11-11
> **Response to Reviewer e76D**
>
> Thank you for spending a lot of time reading the article and giving detailed comments. Responses to some of your questions follow.
>
>
> **The practical value of our work.** We understand your concern that our algorithm does need these assumptions to obtain an unbiased estimate of ATE in theory. However, further supplementary experiments show that our algorithm is still robust and performs better than other baselines even if the assumptions are not satisfied. First, we change the noise terms from Gaussian distribution to uniform distribution that violates Assumption 2. We also change the selection mechanism function with \epsilon*T that violates Assumption 1. The results show that even if the assumptions are not satisfied, the performance of SC-CFR only decreases slightly and still is the best among all methods. And the problem we study is a kind of sample selection bias problem, which is defined in previous work as the outcome data is missing when S=0. In fact, many survey data, such as the job program example we introduce in the introduction, and some application scenarios of the recommendation system. For example, we can collect a large number of user characteristics, but cannot obtain the desired feedback data, i.e., the outcome. Therefore, our method can be applied in a wider range of scenarios and is of practical value. Please refer to the Appendix in the revised version for experimental details and results.
>
>
> **Explanation that the DAG we discuss does not cover all cases of collider bias.** Thank you for your concern. We have also read the book you mentioned. It does contain a lot of more complex cases of collider bias, most of which have some unobserved variables. Since the existence of unobserved variables is more complex and harder to solve, in our paper, we only focus on the situation where confounding and collider bias both exist without unobserved variables. For confounding bias and collider bias with unobserved variables, we will study it in future work.
>
>
> **Some explanations on the experimental results.** First of all, thank you for your careful reading. We have corrected the corresponding clerical errors according to your advice in our revised version. As for the problem that the performance of Direct Estimator improves when sc increases, in fact, with the increase of sc, we can see that the SD of all methods increases, because with the increase of dimensions, the absolute value of y will increase significantly, making the impact of y greater in the sample selection process. Because x is randomly generated, the randomness of S=1 samples will increase eventually. As a result, some extreme situations (for example, sample selection occasionally reduces the impact of confounding bias) are more likely to occur. The reason why CFR and SCCFR have relatively poor effects when sc=10 is that, in order to avoid the uncertainty caused by different hyperparameters in the neural network training process, we use the same hyperparameters in different settings, which may lead to the slow convergence of more complex network structures in low dimensional data.
>
>
> We hope the above answers can solve your problems. Thank you again for your comments and suggestions.

---

### Official Review · Reviewer_CGRS · 2022-10-24

**Confidence:** 4
**Correctness:** 1
**Technical Novelty And Significance:** 2
**Empirical Novelty And Significance:** 2
**Recommendation:** 3

**Clarity, Quality, Novelty And Reproducibility:**

See above for discussion on clarity, quality, and a reproducibility. The contribution appears to be a novel application of recent work on representation learning for causal inference.

**Strength And Weaknesses:**

Strengths:
This submission addresses an important (and challenging) problem, estimating effects in the presence of selection bias.


Weaknesses:
Unfortunately this submission has many problems that must be addressed before publication.

One major problem with this submission is the use of vague and imprecise language in its theoretical claims. For example, in Proposition 2 it is unclear what exactly is meant by "can be calculated and controlled while solving the confounding bias by representation learning". A more precise and falsifiable claim could be an elaborated version of "SC-CFR's output \hat{ATE} is an unbiased estimates of ATE under conditions A,B,C". (As I'll discuss below, the current manuscript does not provide evidence for this specific claim. I am only providing it as a suggestion for future revisions, with appropriate and thorough proof.)

A problem resulting from the above lack of clarity is that the authors have not provided an appropriately rigorous proof of the main results. For example, in Section 3 the authors state that "Since \alpha and \beta are constants, we can consider them as parameters of regression, and thus we can obtain an unbiased estimation of f(X, T) by firstly ..." How can we conclude that the estimator is unbiased? No evidence is provided.

The assumptions used throughout the paper are very strong. While it is fine to make strong assumptions, it is important to provide clarity on their realism, the method's robustness to misspecification, and which (if any) of the assumptions are testable. As an example, assumption 1 is much stronger than an additive noise assumption, as it states exactly how noise in the outcome must be used in the selection assignment mechanism. Note, this is not about how the outcome is used in the selection assignment mechanism, but particularly the outcome noise. It is very challenging for me to think of a scenario where we would want to make such an assumption. Given these concerns, it is surprising that the authors criticize earlier methods for being dependent on strong parametric assumptions. In the related works: "These methods can only solve the simpler case of selection bias caused by covariates or the treatment variable, and suffer from confounding bias in data unless the model specification is correct."

It is also concerning that the paper does not include any supplementary materials. This makes it very challenging to evaluate the empirical results.

**Summary Of The Paper:**

This submission presents Selection Controlled Counterfactual Regression (SC-CFR), a regression-based approach to effect estimation in the presence of observed confounding and selection bias. By making strong parametric restrictions on the relationship between exogenous noise, the selection variable, and the outcome, SC-CFR estimates ATE as the difference between imputed outcomes akin to g-computation. The authors present synthetic results in which the proposed method outperforms baselines.

**Summary Of The Review:**

I am recommending against acceptance on the basis of (1) lack of clarity/precision about mathematical claims and assumptions and (2) lack of evidence for key theoretical claims.

---

> ### Author Response · Authors · 2022-11-11
> **Response to Reviewer CGRS**
>
> Thank you for spending a lot of time reading the article and giving detailed comments. Responses to some of your questions follow.
>
>
> **Problems of imprecise language.** Thank you for your suggestions. We think some of the language problems you pointed out are important, and we have also made some changes to the revised version according to your advice, e.g., We have revised the statement of the proof part of Proposition 2 in Section 3.2 to make it consistent with the content of Proposition 2 itself.
>
>
> **Supplementary explanation and experiments on the assumptions in the paper.** We understand your concern that our algorithm does need these assumptions to obtain an unbiased estimate of ATE in theory. However, further supplementary experiments show that our algorithm is still robust and performs better than other baselines even if the assumptions are not satisfied. First, we change the noise terms from Gaussian distribution to uniform distribution that violates Assumption 2. We also change the selection mechanism
> function with \epsilon*T that violates Assumption 1. The results show that even if the assumptions are not satisfied, the performance of SC-CFR only decreases slightly and still is the best among all methods. Therefore, our method can also be applied in a wider range of scenarios. Please refer to the Appendix in the revised version for experimental details and results.
>
>
> We hope the above answers can solve your problems. Thank you again for your comments and suggestions.

---

### Official Review · Reviewer_xiZL · 2022-10-24

**Confidence:** 4
**Clarity, Quality, Novelty And Reproducibility:** See below.
**Correctness:** 4
**Technical Novelty And Significance:** 2
**Empirical Novelty And Significance:** 2
**Recommendation:** 5

**Strength And Weaknesses:**

See below.

**Summary Of The Paper:**

This paper considers a missing not at random (MNAR) problem. The authors propose a set of parametric assumptions that lead to identifiability. They verify this result on simulated data.

**Summary Of The Review:**

This paper considers the MNAR scenario. This problem has been heavily studied and is known not to be identifiable without making potentially unverifiable assumptions. In this case, the authors make a collection of strong, unverifiable parametric assumptions to achieve identifiability. They verify this approach only in synthetic data that matches these assumptions. I have the following major concerns:

1. My first concern with the paper is that the authors do not make the connection to standard results on missing data or review any of the literature on missing data outside of causal inference. For example, the reason existing approaches focus on settings where $Y \perp S \mid X,A$ is because this is the definition of missing at random (MAR) which *is* identifiable. The authors should include discussion of existing missing data literature and place their work in that context.

2. My second concern is that the assumptions used by the authors to ensure identifiability are strong, unintuitive, and unverifiable. As an example of what can go wrong here, I recommend the authors read Solymos et al. (2012) and the subsequent discussion in Knape & Korner-Nievergelt (2015). Similar to the authors, Solymos et al. use unverifiable parametric assumptions to gain identifiability in a measurement error problem. The problem is that, as Knape & Korner-Nievergelt show, relatively minor deviations from those assumptions can lead to substantial bias. To address this concern, I recommend the following:

a. The authors should provide intuition and examples for their assumptions. Gaussian errors are simple enough, but what does the assumption regarding h imply? Are there known cases that follow these assumptions? If not, why not? Why should we believe these assumptions, in principal?

b. Given that real data will rarely exactly match our assumptions, the authors should examine how sensitive the method is to violations of the assumptions. Currently the synthetic experiments only test the method under the correct assumptions. What happens if, for example, errors are not normal? What if h does not decompose as assumed? See Knape & Korner-Nievergelt for an example of what such sensitivity analyses might look like.

Minor concerns:

1. I think the DAGs in figure 1 over simplify the potential sources of bias. For example, Chapter 8 of Hernan and Robins (2021) consider a host of selection bias and censoring problems that do not match these simple diagrams. I think it is worth clarifying that censoring and selection can happen in a variety of complex ways, not all of which are MNAR.

References:

Solymos, P., Lele, S., and Bayne, E. Conditional likelihood approach for analyzing single visit abundance survey data in the presence of zero inflation and detection error. Environmetrics, 23(2):197–205, 2012.

Knape, J. and Korner-Nievergelt, F. Estimates from non-replicated population surveys rely on critical assumptions. Methods in Ecology and Evolution, 6(3):298–306, 2015.

---

> ### Author Response · Authors · 2022-11-11
> **Response to Reviewer xiZL**
>
> Thank you for spending a lot of time reading the article and giving detailed comments. Responses to some of your questions follow.
>
>
> **Reference to work related to missing data.** We agree with and respect your suggestions. The collider bias we focus on is an MNAR problem, i.e., the outcome data is missing, and the outcome is also one of the causes of sample selection. We have included existing missing data literature in our revised version according to your advice, please see Section 2.
>
>
> **Supplementary explanation and experiments on the assumptions in the paper.** We understand your concern that our algorithm does need these assumptions to obtain an unbiased estimate of ATE in theory. One example of functional forms that meet our assumptions is $s = sigmoid (xt+e^y+\epsilon)$. However, further supplementary experiments show that our algorithm is still robust and performs better than other baselines even if the assumptions are not satisfied. First, we change the noise terms from Gaussian distribution to uniform distribution that violates Assumption 2. We also change the selection mechanism function with $\epsilon \cdot T$ that violates Assumption 1. The results show that even if the assumptions are not satisfied, the performance of SC-CFR only decreases slightly and still is the best among all methods. Please refer to the Appendix in the revised version for experimental details and results.
>
>
> **Explanation that the DAG we discuss does not cover all cases of bias.** Thank you for your concern. As you mentioned, in addition to our DAG, there are other forms of bias, especially in the case of unobserved variables, which may also be confounders or the cause of S. Since the existence of unobserved variables is more complex and harder to solve, in our paper, we only focus on the situation where confounding and collider bias both exist without unobserved variables. However, under the premise that there are no unobserved variables, our method still covers most possible situations, including MAR, for example, only X or T causes S, and so on. For confounding bias and collider bias with unobserved variables, we will study it in future work.
>
>
> We hope the above answers can solve your problems. Thank you again for your comments and suggestions.

---

### Author Response · Authors · 2022-11-11
**Supplementary Explanation and Experiments on the Assumptions in the Paper**

Our algorithm needs two assumptions to obtain an unbiased estimate of ATE in theory. However, further supplementary experiments show that our algorithm is still robust and performs better than other baselines even if the assumptions are not satisfied. First, we change the noise terms from Gaussian distribution to uniform distribution that violates Assumption 2. We also change the selection mechanism function with $\epsilon*T$ that violates Assumption 1. The results show that even if the assumptions are not satisfied, the performance of SC-CFR only decreases slightly and still is the best among all methods. Therefore, our method can also be applied in a wider range of scenarios and is of practical value. Please refer to the Appendix in the revised version for experimental details and results.

---

### Author Response · Authors · 2022-11-11
**On the Scope of the Bias We Discussed**

In addition to the DAG in Figure 1c that we mainly discuss, there are other forms of bias, especially in the case of unobserved variables, which may also be confounders or the cause of S. Since the existence of unobserved variables is more complex and harder to solve, in our paper, we only focus on the situation where confounding and collider bias both exist without unobserved variables. However, under the premise that there are no unobserved variables, our method still covers most possible situations, including MAR, for example, only X or T causes S, and so on. For confounding bias and collider bias with unobserved variables, we will study it in future work.

---

### Decision · Program_Chairs · 2023-01-20

**Decision:**

Reject

**Justification For Why Not Higher Score:**

Reviewers have too many concerns.

**Justification For Why Not Lower Score:**

Overall makes sense, and should be published somewhere eventually.

**Metareview: Summary, Strengths And Weaknesses:**

(a) Provides a method for causal inference that handles both confounding and collider bias, including bias in which outcomes are observed.

(b) Addresses an important problem, sensible method, good experimental results.

(c) Mathematical claims are not precise enough. What does "calculated and controlled" mean exactly?

**Summary Of Ac-Reviewer Meeting:**

No meeting.